# Smart, Photocatalytic and Self-Cleaning Asphalt Mixtures: A Literature Review

**Iran Rocha Segundo** [1,*] , **Elisabete Freitas** [1,*] , **Salmon Landi Jr.** [2,3], **Manuel F. M. Costa** [4,*] and **Joaquim O. Carneiro** [2,*]

1   Department of Civil Engineering, University of Minho, Azurém Campus, 4800-058 Guimarães, Portugal
2   Centre of Physics, Department of Physics, University of Minho, Azurém Campus, 4800-058 Guimarães, Portugal; salmon.landi@ifgoiano.edu.br
3   Federal Institute Goiano, 75901-970 Rio Verde, Goiás, Brazil
4   Centre of Physics, Department of Physics, University of Minho, Gualtar Campus, 4710-057 Braga, Portugal
*   Correspondence: iran_gomes@hotmail.com (I.R.S.); efreitas@civil.uminho.pt (E.F.); mfcosta@fisica.uminho.pt (M.F.M.C.); carneiro@fisica.uminho.pt (J.O.C.)

**Abstract:** Nowadays, there is increasing concern in transportation engineering about the use of techniques less harmful to the environment and also about road safety. Heterogeneous photocatalysis based on the application of semiconductor materials onto asphalt mixtures is a promising technology because it can mitigate air pollution and road accidents. The functionalized asphalt mixtures with photocatalytic capability can degrade pollutants, such as damaging gases and oil/grease adsorbed on their surface, from specific reactions triggered by sunlight photons, providing significant environmental and social benefits. In this article, a review of photocatalysis applied in asphalt mixtures is presented. The most important characteristics related to the functionalization of asphalt mixtures for photocatalytic applications and their corresponding characterization are presented, and the achieved main results are also discussed.

**Keywords:** asphalt mixtures; photocatalysis; optical analysis; smart materials; multifunctional materials; photocatalytic asphalt mixtures; self-cleaning asphalt mixtures

## 1. Introduction

Since the findings of the photocatalytic capability of semiconductor materials, smart products have been gradually inserted in the market during the last three decades [1]. The global market of photocatalytic products is growing from $848 million in 2009 to the expected value of $2.9 billion by 2020 [2]. Nowadays, self-cleaning surfaces based on photocatalytic processes are applied in dissimilar areas such as buildings, road paving, vehicle side-view mirrors, lamps, and even in textiles. Among the semiconductor materials based on oxides, such as titanium dioxide ($TiO_2$), zinc oxide (ZnO) and tungsten oxide ($WO_3$), $TiO_2$ have received the highest attention.

Fujishima researched the photoelectrolysis of water using a $TiO_2$ electrode during the 1960s [3]. Following this revolutionary change in photo-electrochemistry and as a response reaction to the oil crisis of the 1970s, his research pointed to innovative methods for hydrogen production using $TiO_2$ semiconductor material. Nevertheless, due to the need to use ultraviolet (UV) light irradiation (accounting for only about 3% of the solar spectrum) for starting the photocatalysis, the $H_2$ production promoted by using $TiO_2$ appeared to be unattractive. Then, research has pointed out the investigations of $TiO_2$ to air and water-pollutant photodegradation [3].

Road pavements must be able to withstand the effects promoted by vehicle traffic and also by the climate actions (weathering), ensuring driving conditions meet requirements related to safety, comfort,

economy, and with low environmental impact on the surrounding ecosystems. Most recent research on road pavements is intended to improve the mechanical behavior of asphalt mixtures, but recently the functionalization and multifunctional capabilities have become an important topic.

As described by Han et al. [4], a smart and multifunctional cement/asphalt/polymer concrete is a material with properties that differ from conventional concretes, which can react to external stimulus complying with those requirements, such as stress and temperature. These properties can be self-healing, self-sensing, electrically conductive, electromagnetic, and thermal. A smart and multifunctional material is accomplished by proper composition design, special processing, introduction of new functional components, or via the modification of the microstructure from the conventional one [4]. In this sense, a smart asphalt mixture is a material that holds another property besides ensuring good mechanical and superficial performance.

The smart and multifunctional concretes (by their smartness or function), are classified by its smartness, mechanical, electrical, optical, electromagnetic wave/radiation shielding/absorbing, water-related, and energy-harvesting functions. The optical functions are light-transmitting, light-emitting, and photocatalytic ability [4,5].

Heterogeneous photocatalysis mediated by semiconductors has recently attracted significant interest due to its efficient capacity to convert solar energy into chemical energy, mainly in applications devoted to the field of environmental remediation. Several research studies achieved good and promising results related to the degradation of different pollutants emitted by fossil fuels used by road vehicles. Due to the huge surface area of photocatalytic road pavements and its vicinity to the exhaust gases from automobiles, they are quoted as promising surfaces for the reduction (in the presence of sunlight and moisture/$O_2$) of $SO_2$, $NO_x$, $CO_x$, HC, soot fouling and other volatile organic compounds (VOCs) present in the atmosphere [6–13]. Since in the presence of light irradiation, photocatalytic materials can degrade organic pollutants (such as oils and greases) adsorbed to their surface, these materials can also be classified as self-cleaning materials, a very important property in road engineering applications, because the self-cleaning function can contribute to a significant reduction of car accidents on oil-spilled areas.

The most important new functions imparted to asphalt mixtures are the photocatalytic and self-cleaning properties. Therefore, the main goal of this article is to present a review of the functionalization processes performed over asphalt mixtures and also the discussion of their most relevant features. Firstly, the general characteristics of titanium dioxide semiconductor material and its essential principles of photocatalysis will be discussed in Sections 2 and 3, respectively. Afterward, some applications related to civil engineering materials will be presented in Section 4. Concerning the particular applications for asphalt mixtures, Section 5 presents the main materials used and the corresponding published topics, application methods and its benefits and limitations, some surface modifications as a strategy to improve photocatalytic efficiency such as $TiO_2$ metal doping or coupled semiconductor photocatalysts (e.g., g-$C_3N_4$–$TiO_2$ or CdS–$TiO_2$ systems), analysis of results from literature, modeling of photocatalytic pavements, impact on the essential characteristics of asphalt mixtures and application to experimental and real road sections and also their related costs.

## 2. General Aspects and Characteristics of Titanium Dioxide

$TiO_2$ is the most widely used semiconductor material in the field of photocatalysis because of its great ability to promote the degradation of organic pollutants, superhydrophilicity, chemical stability, long-term durability, non-toxicity, visible light transparency [14], low cost and availability (0.44% of the elemental chemical composition of the Earth's crust is Ti) whose worldwide reserves are higher than 600 million tons [15,16].

As with different materials, $TiO_2$ is produced in four different geometric/dimensional shapes (0 to 3). The zero dimensions refer to spherical particles, one dimension is devoted to nanowires, nanorods, nanobelts, and nanotubes, the two-dimensional shapes are for nanosheets, and lastly, the 3D shapes usually refer to porous nanostructures. Obviously, the selection of the geometric/dimensional shapes

depends on the ultimate application purposes [17]. $TiO_2$ nano or microspherical particles are mostly applied in smart materials. They have a high specific surface area as well as high pore volume and pore size, thus increasing organic pollutant adsorption and, consequently, improving the photocatalytic efficiency [14,18].

$TiO_2$ presents three crystalline phases: anatase, rutile, and brookite. Anatase is the most stable crystalline phase, while brookite is uncommon and unstable. Anatase phase can be converted to the rutile phase by heating higher than 700 °C. Anatase and rutile crystallize in the tetragonal system while brookite presents an orthorhombic form. The structure of the anatase phase shows bipyramid tetragonal symmetry with four units per elementary cell, and, for rutile, it is tetragonal, with two formula units per elementary cell [19]. Concerning the photocatalytic ability, the literature reports that $TiO_2$ anatase structure is more efficient than rutile due to its more exposed structure. Degussa P-25 is the trade name of $TiO_2$ particles that are most commonly used in different applications. These particles are composed of about 25% rutile and 75% anatase. This phase composition enables the formation of clusters or thin rutile layers onto the surface of anatase particles, or the presence of individual rutile and anatase nanoparticles, or even heterojunction of the two crystalline phases [20]. The literature refers to the fact that a combination of these two phases can improve photocatalytic efficiency when compared to the use of just one phase [21]. Regarding $TiO_2$ nanoparticles, there are some features that can influence photocatalytic efficiency, namely, their size, specific surface area, exposed surface facets, pore volume and structure as well as crystalline phase content [14].

Besides the photocatalytic properties, $TiO_2$ presents a superhydrophilic surface under photoinduction. A $TiO_2$ surface can change between a less photocatalytic and more superhydrophilic state depending on the chemical composition and processing techniques. When this semiconductor material is irradiated by UV light, the water adsorbed on the surface of $TiO_2$ spreads, forming a thin film, which is related to the reconstruction of hydroxyl groups under UV-light irradiation. Molecular oxygen captures the photo-excited electrons and holes diffuse to the $TiO_2$ surface where they can be trapped by surface lattice oxygen, as suggested by Nakamura et al. [22].

Hole trapping weakens the energy between the Ti atoms and lattice oxygen. Another adsorbed water molecule breaks this bond, forming a new hydroxyl group. The successive dissociative adsorption of water induces the trapping of these hydroxyl species leading to the photogeneration of a hydrophilic domain (size of about 10 nm). Given the unstable state of this surface, the bound photogenerated hydroxyl groups desorb gradually, and the surface returns to its initial state, presenting less hydrophilic behavior [10,23]. $TiO_2$ material's surface with flat or rough texture can exhibit hydrophobic or superhydrophobic behavior when coated with hydrophobic materials, for example, some silane compounds. Moreover, $TiO_2$ surfaces can also present a superhydrophilic state, which, in general, is characterized by a water contact angle less than 5° when irradiated with UV-light [14].

## 3. Basic Principles of Photocatalysis

Photocatalysis is commonly defined as the catalysis of photochemical reactions of adsorbed species on the semiconductor's surface [24].

Unlike metals that have a continuum of electronic states, semiconductor materials possess an electronic structure that is characterized by an empty energy space region where there are no energy levels available to promote the recombination of electron-hole ($e^-/h^+$) pairs generated by photoactivation in the solid semiconductor. The empty energy region is the so-called band gap energy ($E_g$), which extends from the top of the filled valence band (VB) to the bottom of the empty conduction band (CB) of the semiconductor material.

The $TiO_2$ VB mainly consists of oxygen $2p$ orbitals, whereas the CB is essentially formed by the $3d$ orbitals of the $Ti^{4+}$ cations [25].

Since photoexcitation (upon light irradiation) occurs via the band gap, the generated $e^-/h^+$ pairs have enough lifetime, to promote charge transfer to adsorbed species on the semiconductor surface (from the solution or from the gas phase in contact) and participate in redox processes [25].

The interaction of the photogenerated e⁻/h⁺ pairs with chemical species adsorbed to the semiconductor surface generally occurs via two main pathways. At the semiconductor surface, the electrons can be transferred to acceptor species, thus being reduced (usually the electron acceptor is oxygen in an aerated solution); in turn, the holes migrate to the semiconductor surface where they can be combined with donor species (e.g., OH⁻), which become oxidized. Competing with charge transfer to the adsorbed species is the phenomenon of electron-hole recombination. The recombination of the separated e⁻/h⁺ pairs can occur in the bulk of the semiconductor particle or on its surface, releasing energy as light or, more frequently, as heat. Obviously, the recombination of the photoexcited e⁻/h⁺ pairs are detrimental for the photocatalytic performance and, therefore, this phenomenon should be retarded to enable the occurrence of an efficient charge transfer process on the photocatalyst surface. Charge carrier trapping can be used as an important mechanism that can be employed to suppress the recombination process and increase the lifetime of the separated e⁻/h⁺ pairs to participate efficiently in redox reactions. The most effective separation of the e⁻/h⁺ pair is achieved from the Ti defect states and surface Ti–O–Ti sites (or terminal Ti–OH), which can trap the electrons and holes, respectively. In general, oxygen vacancies [26], surface $Ti^{3+}$ [27], Ti interstitial or even ions in the lattice or in the near-surface of $TiO_2$ [28] are responsible by the generation of intra-bandgap energy levels, which are of great importance in retarding the rate of e⁻/h⁺ recombination and simultaneously can enable the absorption of light by $TiO_2$ catalyst, not only in the ultraviolet, but also in the visible region of the electromagnetic spectrum [29–31].

Besides, the e⁻/h⁺ pairs can also be trapped by using other strategies comprising the semiconductor surface modification. For example, in $TiO_2$ photocatalysis, the addition of transition metals (so-called metal doping, for example, iron doping to the semiconductor can change its photocatalytic process by altering the semiconductor surface properties [32]. Upon photoexcitation, the electron migrates to the metal (covering a small area of the semiconductor surface), getting trapped there, and thus, the e⁻/h⁺ recombination is blocked. This occurs due to the creation of the so-called Schottky barrier that is produced at the metal–semiconductor interface, which acts as an effective electron trap avoiding electron-hole recombination and, therefore, increasing the photocatalytic efficiency [33].

Despite this, in several cases related to the preparation of some colloidal and polycrystalline photocatalysts, ideal semiconductors crystal lattices are not formed. Instead, surface and bulk defects occur quite regularly during the photocatalysts preparation process. The nature of surface defect sites depends on the semiconductor's preparation method. A particular example about the role of surface traps is the preparation of CdS colloids obtained by the addition of $H_2S$ to a cadmium salt solution, which generates surface defect sites promoting radiation-less recombination of charge carriers in this semiconductor system. Furthermore, if a surface modification is performed by the addition of excess $Cd^{2+}$ ions and with a proper pH adjustment toward a basic solution, then it induces the blocking of charge carriers trap sites, which also promotes radiation-less recombination of electron-hole pairs across the semiconductor bulk band gap, therefore being detrimental to the photocatalytic performance [33,34].

Figure 1 is a schematic representation of photoexcitation in a semiconductor material in which an electron is moved from its valence band to the conduction band, provided that light is absorbed with energy equal to or higher than the semiconductor's energy band gap.

The general mechanism of photocatalysis using $TiO_2$ under UV light irradiation (wavelength less than 385 nm) follows several stages, which can be generically presented as photochemical reactions shown by Equations (1) to (4) [35–38]. Initially, an e⁻/h⁺ pair is photogenerated by the UV light irradiation on the semiconductor material (Equation (1)). Then, the electrons and holes react with oxygen and moisture, forming super-oxide ions ($O_2^-$) and hydroxyl ($OH^*$) radicals, respectively, which are highly reactive species (Equations (2) and (3), respectively). The $O_2^-$ specie reacts with $H^+$, dissociated from water, and forms $HO_2^*$. From these radicals, pollutant gases such as, for example, $NO_x$, can be degraded, according to Equations (5) and (6) [35–38]. For $NO_x$-type gases, the photocatalysis end products are essentially water-soluble nitrates, which can react by forming mineral salts that are

harmless and very soluble in water [39,40]. In this sense, the end products can be easily removed from surfaces by rain or by simple washing. The soluble nitrates end products should be analyzed to assess their environmental damage. In fact, it is ideally desirable that, after photocatalysis of $NO_x$-type gases, nitrates would be produced with a very low concentration, therefore, without harming the environment.

$$TiO_2 \xrightarrow{h\nu} h^+ + e^- \tag{1}$$

$$e^- + O_2 \rightarrow O_2^- \tag{2}$$

$$h^+ + OH^- \rightarrow OH^* \tag{3}$$

$$H^+ + O_2^- \rightarrow HO_2^* \tag{4}$$

$$NO + HO_2^* \rightarrow NO_2 + OH^* \tag{5}$$

$$NO_2 + OH^* \rightarrow HNO_3 \tag{6}$$

Additionally, the $TiO_2$ semiconductor material can also degrade organic compounds through dissimilar oxidation reactions that, ideally at the last stage, lead to the formation of carbon dioxide and water (Equation (7)). The chemical reaction shown below can be extended to other organic materials such as oil and greases [11], for example, and/or bio microorganisms [41].

$$OH^* + C_nO_mH_{(2n-2m+2)} \rightarrow xCO_2 + yH_2O \tag{7}$$

The understanding of the charge carriers' kinetics and the associated energies of the VB and CB can be accessed from different characterization techniques. For example, diffuse reflectance spectroscopy studies analyzed in light of the Kubelka–Munk theory are commonly utilized to determine the semiconductor's energy band gap [42,43]. Time-resolved spectroscopy and electron spin resonance are applied to monitor the $e^-/h^+$ pairs dynamics and to identify the radicals formed upon the transfer of the charge carriers [44]. Therefore, these results are essential for understanding the mechanisms underlying photocatalysis.

Figure 2a shows the measured values for diffuse reflectance spectroscopy of $TiO_2$ nanoparticles, $TiO_2$ nanoparticles milled at 350 rpm, and $TiO_2$ nanoparticles milled with Fe (at 350 rpm) by using the ball-milling technique. Figure 2b presents the Kubelka–Munk transform, which enables the energy band gap calculation of $TiO_2$ particles.

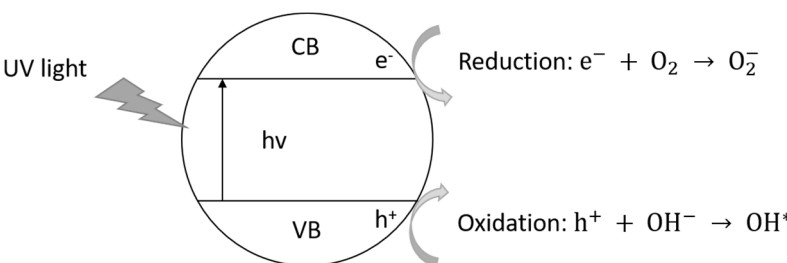

**Figure 1.** Schematic representation of the photo-excitation in a semiconductor material. Adapted with permission from [35]. Copyright 2003 Elsevier.

After acquisition of the $TiO_2$ reflectance curve (reflectance ($R$) versus wavelength), the Kubelka–Munk function can be expressed as $F(R) = (1 − R)^2/2R$ and the Kubelka–Munk transform is given by $[F(R) \times E]^{\frac{1}{2}}$ in which E corresponds to the incident photon energy (see Figure 2b). Taking the curve shown in Figure 2b, the value of the energy band gap was calculated by drawing a line tangent to the curve (drawn from the curve's inflection point) to its intersection point with the energy horizontal axis. Such an intersection point corresponds to the $E_g$ value. For this specific case, the $E_g$ value for $TiO_2$ without milling, milled with and without Fe powder was 3.0, 2.68, and 2.45 eV, respectively.

These results, that is, the changes on the TiO$_2$ energy band-gap, showed that the milling process, as well as doping of the semiconductor by a transition element (iron), influenced the semiconductor's optoelectronic properties [42].

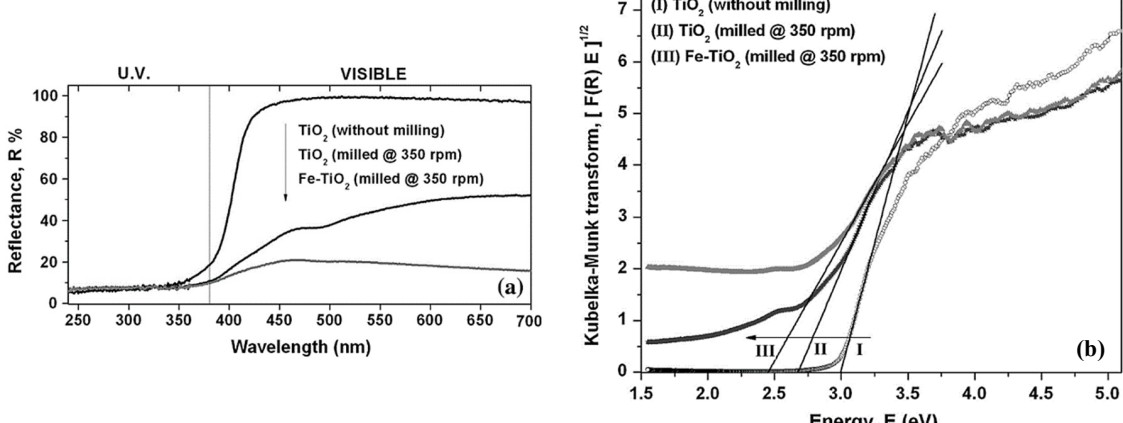

**Figure 2.** Diffuse reflectance spectroscopy of TiO$_2$ nanoparticles (**a**) and calculation of the TiO$_2$ energy band gap by the Kubelka–Munk theory (**b**). Reprinted with permission from [42]. Copyright 2014 Springer.

As mentioned before, the TiO$_2$ semiconductor activates the photocatalysis when it is irradiated by UV light. However, the sunlight is mostly composed of visible and infrared photons, having a very low UV light irradiance, since only about 3%–5% of the solar spectrum comprises the UV range [42]. In this sense, one of the most important concerns reported in literature review to obtain improved photocatalytic materials is the doping of TiO$_2$ particles with different materials in order to decrease its $E_g$ to the wavelength range of visible light (indicated in Figure 2) or forming localized mid-gap states above the valence band [26], thus promoting the degradation of pollutants when subjected to the incidence of photons in the visible range [42,45]. As already pointed out, some TiO$_2$ doping approaches have been studied, such as, for example, via the utilization of transition metals: Cu, Co, Ni, Cr, Mn, Mo, Nb, V, Fe, Ru, Au, Ag, Pt; and non-metals: N, S, C, B, P, I, F [46] or by chemical doping or physical ion-implantation methods [26]. These strategies not only ensure the absorption in the UV range, but also promote the visible-light absorption, thus improving the photocatalytic efficiency under natural solar irradiation [47].

## 4. General Application for Civil Engineering Materials

Semiconductor materials have been extensively used in many applications from common products, such as sunscreens, to advanced and smart materials in the materials science context. The literature refers to the use of semiconductors for chemical (paints), automotive (glasses), aerospace (sensors), photovoltaic (self-cleaning glasses for the optimization of optical transmission), biomedical (functionalization of biomedical prostheses), textile (textiles for treatment of effluents from processing tasks), and civil engineering materials (facade tile materials, glass and pavements) mainly devoted to environmental and energy-related fields [14,48]. The use of the semiconductor materials can consist of the promotion or improvement of some properties, such as self-cleaning, photodegradation of air and water-pollutants, anti-fogging, anti-microbial, anti-aging, cooling surfaces, among others.

In the last few decades, research activity in civil engineering has mainly focused on studying the composition and design of more eco-friendly materials due to the natural resources depletion and environmental damages caused by this activity sector. For the construction of buildings and road pavements, non-renewable sources, i.e., aggregates, cement, and oil products, are required. On the other hand, there has been growing motivation for recycling wastes, the life-cycle study of

different materials, and also their emissions [49,50]. More recently, $TiO_2$ nanoparticles have been incorporated in this sector, for example, in the production of more eco-friendly cementitious and asphalt materials, paints, and finally glasses, which can be devoted to many purposes, mainly for air-cleaning applications.

Ohama and Gemert have divided the use of $TiO_2$ semiconductor in construction materials into 3 different applications: (i) horizontal: concrete pavements, paving block and paving plates, coating systems for pavements and roads, roofing tiles, roofing panels, and cement-based tiles; (ii) vertical: indoor and outdoor paints, cement-based materials, permanent formworks, masonry blocks, sound-absorbing elements for buildings and roof applications, traffic divider elements, street furniture, retaining fair-faced elements; and (iii) tunnels: paints and coatings, concrete panels, concrete pavements, ultrathin white toppings [51].

Industrial activities and road traffic are the main causes of pollutant gas emissions such as $SO_2$, $NO_x$, and VOCs. For example, in France, the health costs only related to road traffic air pollution represent about 0.9%–2.7% of its gross domestic product (GDP) [52]. This type of pollution also reduces the appearance and durability of building materials. The use of inorganic photocatalysts, such as $TiO_2$, has proven to be relatively cheap and an effective strategy to remove toxic organic compounds and pollutant gases from air and aqueous environments. An important and smart application of the $TiO_2$ photocatalyst is concerned with its use in construction and building materials. Moreover, given the building's large surface areas, they can also be used to promote the removal of pollutant gases from the surrounding air [53].

Cementitious materials, especially those used for outdoor applications, are permanently exposed to the action of atmospheric pollutants, microorganisms, and different weather conditions, in which all of them can be related to the deterioration and modification of the material properties. In this context, the use of $TiO_2$ can contribute to increasing the material life cycle while simultaneously promoting air-depollution of polluted areas or the depollution of some specific industries [51,54]. Also, the photocatalytic mortars can remove soot fouling, which deteriorates the architectural appearance of buildings [12].

Besides the self-cleaning effect and photodegradation of air pollutants, $TiO_2$ can be applied to provide passive cooling on the surface of building facades. Under solar light irradiation, $TiO_2$-coated building surfaces become superhydrophilic if subjected to some cooling process, such as spraying water droplets on its surface facades, which spread to form a thin film of water covering large areas of the building's facade. It is important to emphasize that the cooling of a warm building surface (typical case of summer season) is due to a thermodynamic process, which results from the release of heat due to water evaporation when it comes into contact with the building's warm surface. This smart process for cooling down the buildings would decrease the electric energy that is consumed by using conventional air conditioning. Regarding the self-cleaning effect, the use of $TiO_2$ can maintain heritage buildings and also cleaner facades over time [3,18,48]. For this purpose, the pioneering use of $TiO_2$ in Europe took place in the Church Dives in Misericordia, Italy, and in the Music and Arts City Hall in France, which ensured that the original color of their facades was preserved [51].

Another pioneer application of photocatalysis in the civil engineering area refers to the coating of self-cleaning glasses for tunnel lighting in Japan. Uncoated lamps tend to lose brightness due to contaminants from vehicle exhaust gases that are adsorbed onto the lamp exterior. Sodium lamp glasses coated with $TiO_2$ semiconductor emit enough UV light ($\sim$3 mW/cm$^2$) to enable catalytic reactions and photodecompose adsorbed contaminants [10].

For indoor applications, $TiO_2$ coated ceramic tiles, and mortars are considered to be very effective against bacteria [55] and also fungi [56]. The bacteria are killed faster than they can grow on photocatalytic ceramic tiles. This application in hospitals and health centers can be used to reduce the spread of infections. Moreover, it can be used for public, commercial, and residential buildings to improve the hygienic conditions [55]. Considering the antifungal application on mortars, Jerónimo et al. evaluated the growth evolution of the *Cladosporium* fungus. They found that after 49 days, in the case

of the mortar without $TiO_2$, the fungus began to develop while the one with 4% of $TiO_2$, the fungus was still practically germinating [56].

The $TiO_2$ semiconductor material has also been applied on (window) glasses for anti-fogging and self-cleaning purposes [57,58]. The use of self-cleaning properties on glass windows can save money on building maintenance budgets, like cleaning by using conventional methods which is expensive and laborious, especially when dealing with high-rise buildings. Its main benefit is the combination of hydrophilicity and pollutant photodegradation, which significantly helps the cleaning process [58]. The next section presents additional aspects related to the application of $TiO_2$ on asphalt pavements, especially those related to de-pollution purposes.

## 5. Photocatalytic Asphalt Pavements

Road pavements are ideal places for reducing air pollution due to their large paved surface areas, which are in close vicinity with the toxic gases emitted by the vehicles circulating there [59]. Figure 3 schematizes the operational mechanism of the asphalt mixtures with photocatalytic ability, one of the most evaluated characteristics in multifunctional road pavements.

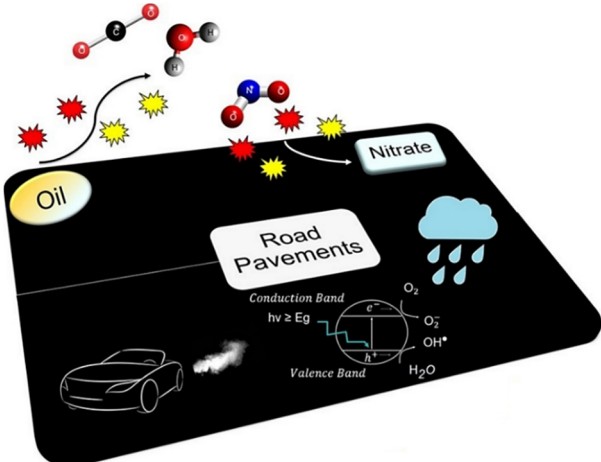

**Figure 3.** Photocatalytic asphalt mixture. Reprinted with permission from [8] Copyright 2018 Elsevier.

Photocatalytic surfaces are also considered self-cleaning due to their ability to degrade organic pollutants, such as adsorbed oils and greases. In a road safety context, the ability to degrade organic compounds adsorbed on paved surfaces can prevent problems related to vehicle skidding, thus contributing positively to the reduction of road accidents [6].

The number of articles published on topics related to photocatalytic asphalt mixtures has been steadily increasing. When looking for the keywords "photocatalytic" and "asphalt mixtures" (using Google Scholar), it is possible to observe that the research about this topic rose from 2 to 40 occurrences between 2006 and 2018, respectively, excluding citations. When carrying out the same analysis but only searching the keyword "asphalt mixtures", it is possible to count the total occurrences of 904 and 3220 for the same period, respectively. This number means that in 2018, only 1% of occurrences refer to research about the topic of photocatalytic asphalt mixing. On the other hand, for this research topic, the number is increased by about 65% when the word "mechanical" is included in "asphalt mixtures", thus showing that the mechanical behavior of asphalt mixtures is the most important issue related to asphalt pavements. Moreover, it can be concluded that the research in photocatalysis applied to asphalt mixtures is quite innovative and is of great interest to researchers.

### 5.1. Materials and Techniques for Application of Photocatalytic Effect on Asphalt Mixtures

Like many other applications in the field of material science, the most important semiconductor material in photocatalytic asphalt mixtures also refers to $TiO_2$ [6,60,61]. Meanwhile, the literature

also presents information regarding the use of other materials, such as ZnO [7] and graphitic carbon nitride (g-C$_3$N$_4$) [62,63]. ZnO is also a semiconductor material with a band gap of about 3.3 eV [64,65], while g-C$_3$N$_4$ is a new type of non-metal photocatalyst that has a band gap of ~2.7 eV [62,63,66]. TiO$_2$ is commonly used on a nanometer scale (between 6 and 40 nm) [38,67–69]. In photocatalytic asphalt mixtures using TiO$_2$, the crystalline phase of anatase [9,67,68,70,71] and the TiO$_2$ nanoparticles P-25 [6,7,61] are commonly used. Analyzing the photocatalytic capacity applied to asphalt materials, quite a few works can be found on conventional asphalt mixtures [6,59], asphalt mixtures with lower production temperature (warm mix asphalt) [70], slurry seals [62], fog seals [72], asphalt emulsions [63,73], among others. Also, other issues have been studied, such as their application in real context or just small test sections [61,68,69,71,73,74], and the computational modeling of the photocatalytic capability of road pavements [74,75].

In order to obtain photocatalytic asphalt mixtures, it is possible to highlight four main techniques that can be utilized to apply the semiconductor materials in the asphalt mixtures: (i) spraying coating [6,9,39,45,67,70,72], (ii) volume incorporation [6,67,72,76], (iii) bitumen modification [70], and (iv) spreading [67,72,77,78]. Bitumen modification is developed before the manufacture of the asphalt mixture. The semiconductor particles are mixed just with the binder. The technique by volume incorporation is carried out during the asphalt mixture production where the particles are inserted and combined with the aggregates and/or the filler. The spraying coating and the spreading processes are applied over the asphalt mixture after its compaction. The first one, that is, the spraying coating, is carried out using a spray-painting gun and the second one (spreading process) is carried out by using a specific solution that is deposited over the surface like a chip sealing or even using paintbrushes. There are other techniques that can be used to functionalize asphalt mixtures, but probably implying a higher degree of application difficulty and also leading to a higher cost of implementation in a real context, such as the use of the atmospheric plasma-spraying technique [69].

Spraying coating is probably the most efficient functionalization technique, and it also uses smaller amounts of semiconductor material. However, the immobilization of the semiconductor particles over the asphalt mixtures surface is still a major challenge. The second and third functionalization processes, i.e., the volume incorporation and bitumen modification, respectively, probably guarantee the best immobilization characteristics, but more material needs to be used compared to the other ones. Nevertheless, traffic abrasion is required to expose the semiconductor material that is covered by bitumen.

The modification technique improves the aging resistance of asphalt binders mainly due to the capability of the semiconductors to reflect and absorb the UV light, which is the most important issue for the long-term aging process [70,79–81]. Additionally, TiO$_2$ nanoparticles as bitumen modifier decrease the acid component and increase the alkaline component and also the surface-free energy of the asphalt binder, leading to improvements in the adhesion between the asphalt binder and the aggregate [82].

Regarding the spraying coating technique, the literature reports the application of aqueous solutions [6,9,39,70,72] and other ones with emulsions [73], cement and resins [38,77], polymers [83] and rubber [68] in order to achieve a better immobilization process. A significant concern that should be taken into account when it is desired to obtain photocatalytic aqueous solutions is the dispersion of the TiO$_2$ nanoparticles; otherwise, they may agglomerate and, consequently, decrease the photocatalytic efficiency. Some authors submitted TiO$_2$ solutions to ultrasonic treatment [45,67] or prepared aqueous solutions with a value of zeta potential different from the isoelectric point [6]. The previous preparation reduces particle agglomeration because of the van der Waals attraction forces that should be competing with the Coulomb's repulsive electrostatic forces [6]. For the preparation of the TiO$_2$ aqueous solution, the suggestion is to use a pH equal to 8 [6].

To improve the photocatalytic efficiency and the immobilization of the nanoparticles, Leng and Yu proposed an interesting spraying technique called Breath Figure (BF). The process consisted in a solution with TiO$_2$ nanoparticles mixed with asphalt binder, polymer (polystyrene) and solvent

(tetrahydrofuran). This process can create porous microstructures in bitumen, thus creating suitable conditions for $TiO_2$ particles to enter and be deposited inside the pores. The photocatalytic efficiency of the BF asphalt mixture was higher than that obtained with the asphalt mixture sprayed with $TiO_2$ aqueous solution before and after abrasion condition. The best proportion was 2:1:1 of $TiO_2$, binder, and polymer, respectively, and the concentration of all solutes was 120 mg/mL [83].

For the case of asphalt mixtures, the application of $TiO_2$ doping by using dissimilar materials, the literature reports the its functionalization by La [45], Ce [67,84], Cu and Fe by ultrasonic-assisted sol-gel [84], N by simple mixing [85], and C [61] and Ag [68]. The authors concluded that $TiO_2$ doped with 0.5% of La (mol%) [45] and 0.2% of Ce (mass ratio) [67] presented the optimal photocatalytic efficiency. Chen et al. mixed $TiO_2$ and urea at a 2:1 ratio in molar proportions [85]. Unfortunately, neither C nor Ag content was reported by Fan et al. [61] and Liu et al. [68], respectively. Under visible light, all of these treatments were able to degrade pollutants (such as $NO_x$) and improved photocatalytic efficiency because, considering the solar spectrum on Earth, visible light irradiance is much higher than UV. Other materials have been added to $TiO_2$ in order to improve its photocatalytic efficiency or even to improve another capability, such as the superhydrophobic ability. The non-metal g-$C_3N_4$ improved the NO degradation [63] and ZnO increased the water contact angle of the asphalt mixtures [7] when they are combined with the $TiO_2$ semiconductor material.

Considering the UV light resistance of asphalt binders modified with $TiO_2$ doped with Cu, Fe, and Ce (molar mass ratio of $[Ti^{4+}$/Metal ion] equals to 3/1), Jin et al. [84] concluded that the optimal content of these materials is 5% (in the binder mass). Comparing the results obtained, the best performance was achieved by using Ce. The photocatalytic efficiency increased by at least 27% for NO and 40% for HC, after 90 min of testing, when the $TiO_2$ was doped when compared to the samples with only $TiO_2$. Regarding the HC and NO degradation, the best doping materials, presented in ascending order, are Cu, Ce, and Fe [84].

*5.2. Testing, Results Analysis of Photocatalytic Asphalt Mixtures*

In order to evaluate the photocatalytic ability, the literature presents gas degradation tests, such as $NO_x$ and VOCs, most of them according to the standards ISO 22197-1 and JIS TR Z 0018 [60,67,70], and tests of photocatalytic efficiency via the degradation of different organic dyes, such as methylene orange (MO) [61], methylene blue (MB) [6,67] and rhodamine B (RhB) [7,8], which use spectrophotometric techniques (Figure 4). Both tests can be used to evaluate the photocatalytic gas degradation, considering both air depollution and dye degradation for self-cleaning applications. Nevertheless, both tests are clearly related as they are used for measuring the ability of pollutant photodegradation.

Regarding the first test, some gases have been used: some VOCs: HC [84], benzene, toluene, xylene and trimethylbenzene [59], $SO_2$ [9] and mainly NO, $NO_2$ or $NO_x$ [68,71,83]. Briefly, the gas flow with a controlled concentration passes into a chamber where the sample containing the semiconductor material resides inside and is subjected to light irradiation (Figure 4a). The gas concentration is monitored at the outlet of the system and the photocatalytic efficiency can be further evaluated. It is possible to control humidity, temperature, light irradiation (wavelength and power), gas flow rate and concentration, and other physical parameters.

Regarding the second test, that is, the organic dye degradation, the asphalt mixture samples under study are dipped into a dye's aqueous solution of known initial concentration. Subsequently, the sample already immersed in the dye's aqueous solution is exposed to light irradiation and, over time, the variation in solution concentration is monitored by spectrophotometry (Figure 4b,c). The photocatalytic efficiency can be calculated from detected variations (decrease) in the maximum absorbance values recorded on the absorption spectrum by using visible spectroscopy. As before, it is also possible to control light irradiation (wavelength and power), the dye's solution concentration, volume, and pH.

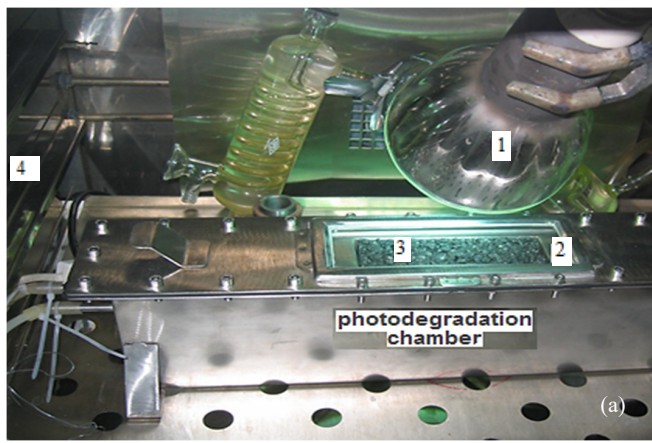
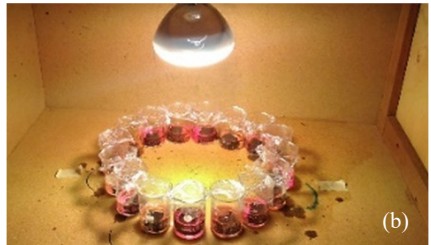
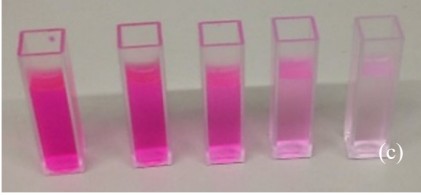

**Figure 4.** Experimental tests to measure the photocatalytic activity: (**a**) $NO_x$ degradation efficiency (1: lamp for light irradiation, 2: quartz window, 3: test specimen, and 4: temperature stabilizer); (**b**) degradation of an organic dye; (**c**) aqueous solution of an organic dye of different color intensity over time, thus indicating its degradation.

Figure 5 shows the results related to the two types of tests mentioned previously. For the gas degradation (Figure 5a), a plot of gas concentration, in this case, $NO_x$, versus duration (degradation time) is presented [60,86]. During the test beginning, the UV light is turned off. After reaching the equilibrium condition (after about 1 h 45 min), the UV-light source is turned on. Next, the gas is released, flowing into the chamber, and it is observed that the NO gas concentration decreases over time, thus showing a photodegradation capability. Actually, after 5 h of testing, the NO gas concentration becomes essentially constant, showing a reduction in its concentration of 72 ppb, which is a value that corresponds to about 18.8% when compared to the initial gas concentration (it was about 390 ppb). After 7 h of testing, the gas flow and the UV light source were turned off.

Figure 5b shows the absorbance spectra of the methylene blue aqueous solution acquired for different UV light irradiation times [6]. For the calculation of photocatalytic efficiency, the decrease of the maximum absorbance (wavelength equals to 662 nm) is observed. Using the Beer–Lambert law and considering that the molecular extinction coefficient and the light path length are constant, the concentration and the absorbance are directly proportional (Equation (8)), and the photocatalytic efficiency can be calculated from Equation (9). It is observed that the sample's photocatalytic efficiency increases over time of photo-irradiation, because the absorbance of the dye's aqueous solution decreases (or similarly, its concentration decreases over time), thus indicating the occurrence of the photodegradation process. In this specific case, the tests were conducted for 480 min. Thus, it is possible to present a plot of photocatalytic efficiency versus photo-irradiation time. It is also important to emphasize that some details can be carried out to better evaluate the photocatalytic ability, thus avoiding some errors or misunderstandings. When the samples are immersed in the aqueous dye solution, still without UV light irradiation, a first phenomenon occurs, which is the adsorption of the dye onto the sample's surface. Therefore, it is recommended that samples should be kept in the dark until the adsorption–desorption equilibrium is reached [87]. Additionally, due to the granitic nature of the aggregates, some photocatalytic efficiency also occurs for blank samples, that is, those that have not been modified by the addition of semiconductor material. In order to avoid or even eliminate the side effect of exposed aggregates, the side and bottom faces of the samples should be coated with bitumen, thus only showing their upper face that is the traditional condition. If this procedure is not performed, this must be considered for the results, that is, photocatalytic efficiency for the blank samples [8].

$$\frac{A}{A_0} = \frac{C}{C_0} \tag{8}$$

$$\Phi\,(\%) = \left(\frac{A_0 - A}{A_0}\right) \times 100 \qquad (9)$$

where $A$ and $A_0$ are the maximum absorbance of the dye's aqueous solution for a determined instant of time, $t$ and for initial time $t = 0$ h, respectively; meanwhile, $C$ and $C_0$ represent the concentrations of the dye's aqueous solution for the same instants of time mentioned previously and $\Phi$ is the photocatalytic efficiency.

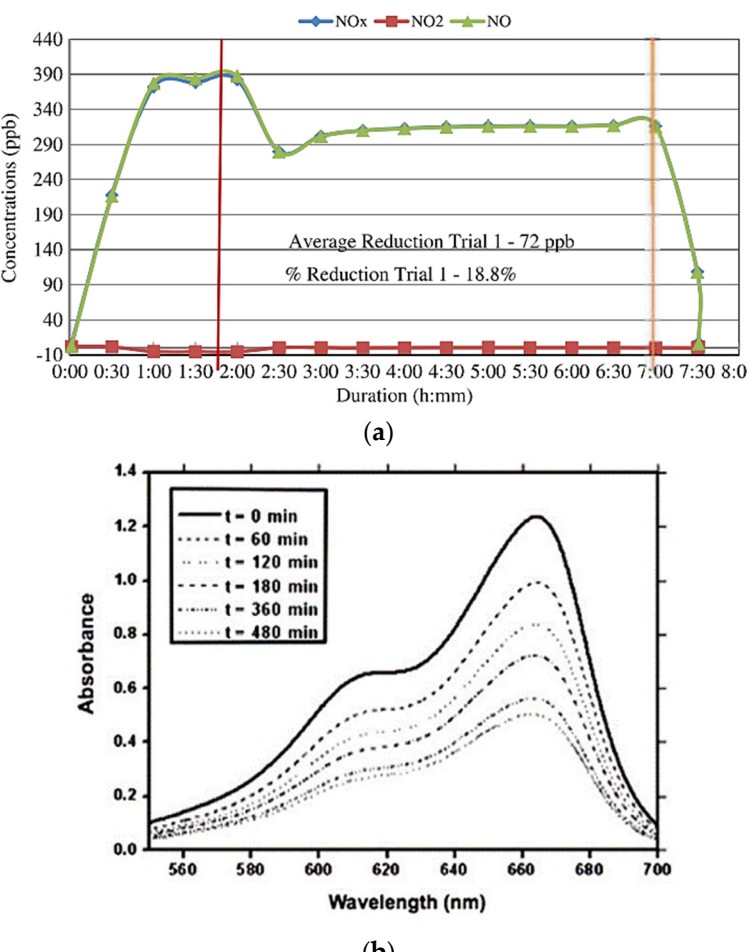

**(a)**

**(b)**

**Figure 5.** Results of photocatalytic tests: (**a**) variation (over time) of gas concentration, indicating its photodegradation (Reprinted with permission from [60] Copyright 2014 American Society of Civil Engineers); (**b**) absorbance variation (over time) of the dye's aqueous solution, indicating its photodegradation (Reprinted with permission from [6] Copyright 2013 Elsevier).

In these tests, the source light used to promote the samples irradiation may be only the UV light (shorter wavelengths of about 300 nm) [6] or simulation of sunlight spectra [7,8] and also visible light (for example, by using a halogen lamp with a wavelength of 400–800 nm or light-emitting diodes (LEDs) in different wavelengths) [45,67,85]. The first option, that is, the use of UV light irradiation takes into account only the photocatalytic process activated in this region of the electromagnetic spectrum. The irradiation that simulates the sunlight considers real situations of outdoor applications. The last one, i.e., the samples irradiation over the visible range of the electromagnetic spectrum, is carried out in order to study the effect of semiconductor doping on the photocatalytic efficiency. For example, in order to evaluate the photocatalytic efficiency of doped-TiO$_2$ coated asphalt mixtures, Chen et al. used lamps operating in different wavelengths, namely 330–420 nm, 430–530 nm (blue LED), 470–570 nm (green LED), and 590–680 nm (red LED). They have concluded that the N-doped TiO$_2$ asphalt mixture

presented a better photocatalytic efficiency than the conventional $TiO_2$-coated asphalt mixture. Besides, the photocatalytic efficiency decreases with the increase of the wavelength [85].

Another technique, proposed by Osborn et al., consists of adding water over the photocatalytic surface and, after 5 min, collecting this aqueous solution with a syringe to analyze the number of existing nitrates [9,60]. This technique resulted in 48% of the nitrate removal when compared to the standard method, that is, the ISO 22197-1 standard. It is a simple procedure that does not require the acquisition of asphalt samples from the field.

The oil/grease degradation could also be analyzed. For example, instead of evaluating the photocatalytic efficiency through the degradation of some organic compounds (dyes or soot) [6,13], there is the opportunity to evaluate this ability via the degradation of some simple oil/grease compounds that can be analyzed by spectrophotometry, for instance.

The photocatalytic efficiency can be influenced by several factors, such as pollutant flow rate and concentration, solar irradiation, humidity, environmental temperature, among others. Due to the action of traffic and wind speed on the photocatalytic asphalt mixture, a decrease in its photocatalytic efficiency in $NO_x$ degradation usually occurs by about 60% and 42%, respectively [71]. An increase of the solar irradiation also leads to the concomitant increase in photocatalytic efficiency. The presence of excessive water inhibits the photocatalytic reaction as humidity competes with pollutants for adsorption over the surface, thus decreasing the photocatalytic efficiency [9,69,71]. On the other hand, the increase of gas flow rate decreases the photocatalytic efficiency due to an insufficient photocatalytic reaction under high-speed gas flow, because it reduces the contact time between the reactants and the catalyst [9,45].

Moreover, the gas concentration also influences the photocatalytic efficiency. The higher the concentration, the lower the photocatalytic efficiency [45,68]. Light irradiation with shorter wavelengths is more effective than for longer wavelengths [68,69]. A decrease in photocatalytic efficiency when the samples are submitted to higher temperatures has also been observed, which can be explained by the higher kinetic energy of the molecules, thus accelerating their gasification and reducing the contact of the gas with the semiconductor material [68]. Additionally, the dust particles over the asphalt mixtures also contribute to reducing the photocatalytic efficiency [76].

Besides the photocatalytic effect, this technology promotes a temperature decrease in the urban environment. Due to the color and light reflection, the $TiO_2$ treatment over the asphalt mixtures surface can reduce their surface temperature. This application can mitigate urban heating island (UHI) and building cooling demand [88]. The registered maximum temperature decrease, in a summer day, was 5.5 °C for a cool off-white asphalt when compared to the conventional asphalt mixture.

## 5.3. Photocatalytic Asphalt Mixtures Modeling

Some authors developed models related to the NO degradation promoted by photocatalytic asphalt pavements. Equations (10) to (12) describe NO concentration by factors related to the number of vehicles per hour ($T$), relative humidity ($H$), wind speed ($V$) in m/s, ambient temperature ($T_{Ambient}$) in °C and solar radiation rate ($S$) in $W/m^2$ [74].

$$NO_{before} = 0.96 \times T + 0.22 \times H - 1.33 \times V_{Ambient} - 10.5 \times V + 0.02 \times S \tag{10}$$

$$NO_{after} = 0.31 \times T + 0.06 \times H - 0.1 \times T_{Ambient} - 0.75 \times V + 0.0003 \times S \tag{11}$$

$$NO_{reduction} = NO_{before} - NO_{after} \tag{12}$$

where $NO_{before}$ and $NO_{after}$ correspond to the hourly average of NO (ppb) concentration before and after the application of $TiO_2$, respectively and the $NO_{reduction}$ corresponds to the difference between these values. The modeling presented a $R^2$ of 0.79 for the untreated samples and 0.67 for the treated ones. As there are just a few studies about the photocatalytic efficiency of asphalt mixtures, therefore this must be better evaluated.

Kruschwitz et al. [75] report a 3D simulation of the whole length of photocatalytic asphalt pavement, showing similar results between the finite-element simulation technique and the experimental measurements of NO vs. time. They present some variables, such as the initial concentration of NO, vehicle distribution, NO emission by vehicle, road geometry, solar effects on NO (natural conversion from NO to $NO_2$), and pollution-reducing effects. They also showed uncertainties in the weather condition (for example, UV irradiation, wind, and rain) [75].

### 5.4. Mechanical and Surface Impacts and Design of Photocatalytic Asphalt Mixtures

The mechanical characterization of photocatalytic asphalt mixtures has already been carried out. By volume incorporation, $TiO_2$ leads to a decrease in moisture resistance, while fatigue and rutting may also be affected. A high content of $TiO_2$ (6%) improved the rutting resistance but decreased the fatigue resistance, whereas a low content of $TiO_2$ (3%) slightly increased the rutting but maintained the fatigue resistance [8]. Furthermore, the functional impact was also evaluated. The functionalization techniques by spraying coating using aqueous solution and volume incorporation led to a difference in friction between −7% and 3%. The microtexture amplitude parameters were not affected by any functionalization technique except the skewness of AC 14 with 6% $TiO_2$ by volume incorporation [89].

Wang et al. [78] concluded that the friction and texture depth of the asphalt mixtures decreased when the spreading $TiO_2$ coverage rate (by emulsion spreading) is increased. Considering the local technical requirements, $550 \ g/m^2$ is the maximum coverage rate that respects these parameters. The same authors studied the water permeability of the photocatalytic asphalt mixtures. They concluded that the covering rate did not affect the water permeability. They recommended 8% $TiO_2$ with a covering rate of $400 \ g/m^2$.

A very interesting article was published taking into account the smart asphalt mixture design. Analyzing a mortar composed of asphalt binder, filler, and $TiO_2$, Zhang et al. [76] showed that the stiffness modulus was not influenced by the $TiO_2$ content. The increase of $TiO_2$ content raises the cone penetration values at 60 °C, which can indicate lesser performance of permanent deformation resistance. Analyzing the mechanical behavior of the asphalt mixture functionalized by volume incorporation, they concluded that the $TiO_2$ content did not affect the water sensitivity and the low-temperature anti-cracking performance, but it affects the permanent deformation. Considering the maximum value for the deformation, they reported that the maximum content of $TiO_2$ that respects the mechanical behavior of the asphalt mixture is 4%. Regarding the accumulative decomposition ratio of automobile exhaust degradation, they reported that the best content of $TiO_2$ is 3.1% in mass of the aggregates by volume incorporation. Considering the voids content of the asphalt mixture, the efficiency increases with the increase of voids content because of the higher contact area between automobile exhaust and asphalt mixtures. Considering the compaction, they observed that increasing in the number of gyrations by the Superpave Compactor leads to the decrease of the photocatalytic efficiency. They suggest the application of an asphalt mixture with high air voids content, for example porous asphalt concrete. Thus, obviously, during the service period of the photocatalytic asphalt pavement, the photocatalytic efficiency decreases due to the compaction.

These concerns consist of the design of smart asphalt mixtures taking into account the essential parameters (mechanical and superficial performance) and the new capabilities. For coatings by spreading or spraying over asphalt mixtures, the functional (superficial) characteristics should be respected. For the volume incorporation, besides the functional characteristics, the mechanical performance should be assessed. Last but not least, those functionalized by bitumen modification must be evaluated under aging resistance, mainly by UV light. Under these circumstances, it is possible to obtain the optimization of the new capabilities (in terms of content, covering rate, etc.), respecting the standard requirements for real applications.

### 5.5. Photocatalytic Applications in Real Context or Small Road Sections

Regarding the application in real scenario context or small test sections, some first approaches were performed (Figure 6). Bocci et al. applied a bituminous emulsion containing $TiO_2$ and cement mortar made of sand, cement, additives, and $TiO_2$ sprayed over asphalt mixtures onto Highway A14 in Italy. The coatings were applied to the right and emergency lanes of the road and their performance was monitored for 527 days. The most important variables that contributed to the decrease in emulsion efficiency were the low temperatures and rainfall. Its efficiency has been reduced by about 80% in just 100 days. The authors concluded that, for tunnel applications, where weathering would be avoided, the best alternative should consist of the application of a bituminous emulsion, while, for the case of emergency lanes, the best option could fall on the use of cement mortar [73].

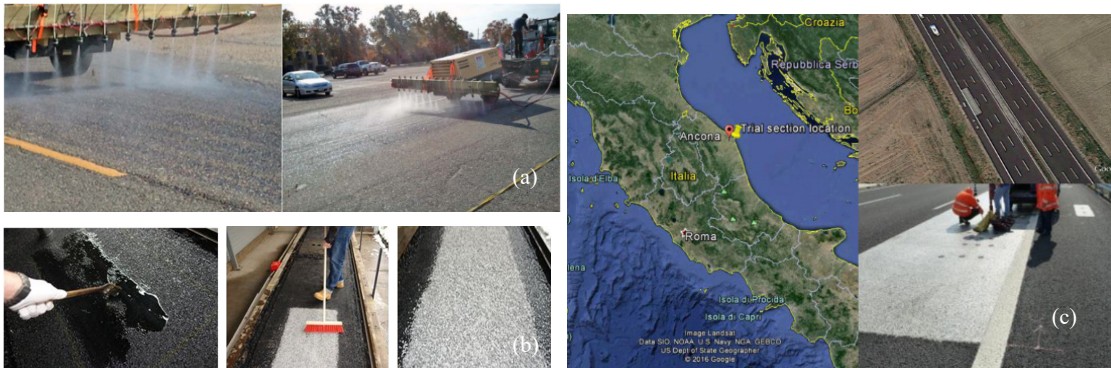

**Figure 6.** Photocatalytic test sections: (**a**) by Hassan et al. (Reprinted with permission from [9] Copyright 2012 Elsevier); (**b**) by Wang et al. (Reprinted with permission from [77] Copyright 2017 Elsevier), and (**c**) by Bocci et al. (Reprinted with permission from [73] Copyright 2016 Springer).

Meanwhile, Lui et al. applied a composite material (composed by Ag-doped $TiO_2$ nanoparticles dispersed in silane-bound rubber powders and dissolved in ethanol) to coat the asphalt mixture surface layer of the Wenxing Tunnel in Xiamen, China. According to their study, the results obtained showed that when the technology was applied to zones in the middle of the tunnel, a $NO_x$ gas uptake of at least 62.4% was registered. When this result was compared to that obtained for tunnel exit sites, they found a decrease in gas uptake by about 26.3% because of the easier gas exit from the tunnel (i.e., at the tunnel end sites, the gas is closer to the outside) [68].

Moreover, Chen and Liu also applied a photocatalytic coating composed of $TiO_2$ nanoparticles dispersed in silane on the asphalt mixture of Highway G11 from Tsitsihar to the Nehe River, China. They showed a photocatalytic efficiency from 6% to 12% under real scenery of outdoor road traffic [69].

Wang et al. [77] spread cement mortars containing $TiO_2$ particles bound by epoxy resin in asphalt pavements of a test section at the Institute of Highway Engineering at Aachen, Germany. The authors observed that, without using the polishing process, the technology presented a photodegradation capacity of about 25.2%, while after 300 min of the polishing task (equivalent to a period of 8–15 years of traffic), the samples showed a photocatalytic efficiency of about 10.7%.

Fan et al. [61] sprayed a C-$TiO_2$ aqueous solution over a 90 $m^2$ section of asphalt road in Sheung Shui District, New Territories, Hong Kong. After applying the treatment, they heated the asphalt road surface by using an infrared heating lamp in order to immobilize the C-$TiO_2$ nanoparticles and concluded that heating treatment was successful. Besides out-of-the-wheel tracks, the nanomaterials remained since there is no traffic wearing [61].

Another study suggests that the durability (permanence) of $TiO_2$ nanoparticles sprayed over asphalt pavement surfaces ranges from 10 to 16 months [60]. Recently, Chen et al. applied a spraying coating of N-doped $TiO_2$ (by using a silane coupling reagent) over an asphalt road and showed that the coating's durability kept working during about 13 months [85].

The literature points out that the application of photocatalytic treatments on asphalt mixtures is still being studied. There are just a few articles published about the analysis in a real context. In fact, ensuring the nanoparticles immobilization over surfaces still remains the major problem and challenge for researchers. The main and most commonly used application techniques are those that interact with the surface: spreading and mainly spraying coating. The others, that is, the bitumen modification and volume incorporation techniques, should be tested since there is a lack of information on these approaches under test section or real context applications.

*5.6. Cost Analysis*

Regarding the costs, probably between the known four application methods, those that use the least amount of semiconductor material and also is more efficient, refers to the spraying coating technique. Hassan et al. carried out an economic analysis of a photocatalytic coating of $TiO_2$ aqueous solution sprayed over the asphalt pavement surface layer. This solution was composed of anatase phase of $TiO_2$ nanoparticles (2% by volume), and it was applied covering 0.05 $L/m^2$. They concluded that the photocatalytic coating costs were about USD 2.25 per $m^2$ (in 2012). When compared to the application cost around 9 cm asphalt mixture layer, the coating application increased by 11% the initial budget [9]. Fan et al. reported that the fabrication cost of the $C-TiO_2$ aqueous solution coating applied by spraying is estimated in USD 1–3 per $m^2$ covering about 7 $g/m^2$ (in 2018), but the other costs need to be evaluated in further studies. Bocci et al. studied bituminous emulsion containing $TiO_2$ and cement mortar made of sand, cement, additives, and $TiO_2$ sprayed over asphalt mixtures. They reported a cost of €6.50 per $m^2$ (0,067 $kg/m^2$) for the first one and €20 per $m^2$ (1 $kg/m^2$ except for water) for the second one, in 2015.

As a typical example related to the costs of raw $TiO_2$ nanoparticles, one can highlight the case of Evonik Aeroxide P-25, which presents a cost of about USD 45 per kg [90]. Nevertheless, it is well known that $TiO_2$ nanoparticles are still quite expensive due to very low industrial production. It is predictable that, in the next years, this cost would decrease significantly due to the very high application versatility of $TiO_2$ nanoparticles. Besides, since these researchers report very small test sections, the application over asphalt roads in a real context that covers much larger areas necessarily requires the acquisition of a large amount of nanomaterials, thus leading to a decrease in the nanoparticles' price. Because there is a lack of information about the cost analysis, this approach could be evaluated by researchers in further studies.

It can be concluded that photocatalytic road pavements are very important since if they are close to the largest emissions, they have a large area, and the pavements are mostly bituminous. As mentioned, there are important references related to the functionalization of asphalt mixtures to impart photocatalytic function. However, there is still limited experience on this topic, thus the surface immobilization techniques of semiconductor materials need further studies. Other aspects also require further investigation, such as the application of semiconductor materials in different substrates (asphalt mixtures), the impact of functionalization on the functional and mechanical characteristics, application costs, and real application analysis.

## 6. Conclusions

Photocatalytic asphalt mixtures behave like smart materials since they present new capabilities when subjected to the action of light (optical interaction). Photocatalysis applied in this field is very innovative and is of great interest to researchers. This new capability is mainly provided by the presence of semiconductor materials, mostly $TiO_2$, usually at the nanometer scale (anatase phase or P-25) and applied by using four major methods: spraying coating, bitumen modification, volume incorporation and spreading.

When irradiated by UV light from sunlight, the $TiO_2$ semiconductor material activates the photocatalysis process. Nevertheless, the sunlight is composed of only 3%–5% of the UV range light. Therefore, many research efforts have been carried out in order to shift the $TiO_2$ absorption energy to

the visible range of the electromagnetic spectrum; a common strategy consists of using a $TiO_2$-doping process with different chemical elements in order to obtain an improvement in the photocatalytic efficiency. Under the scope of road pavements, the photocatalytic efficiency can be evaluated by degradation of a harmful gas (mainly $NO_x$) and monitoring over time the variation of the concentration of a particular organic dye's aqueous solution (MB, MO or RhB).

There exist some variables that can influence the photocatalytic efficiency, such as pollutant flow rate and concentration, solar irradiation, humidity, and environmental temperature. In general, a high concentration of pollutants, high environmental temperature, high relative humidity, high gas flow rate, high traffic, and wind speed contribute to decreasing the photocatalytic efficiency. On the other hand, a high level of solar irradiation increases photocatalytic efficiency. In addition, the incidence of higher energy photons on a semiconductor material becomes more effective than when the incidence occurs with less energetic photons (longer wavelengths). Additionally, higher asphalt mixture air voids and cleaner surfaces improve the photocatalytic efficiency.

In order to design a smart asphalt mixture, the researchers must consider the essential characteristics (mechanical and superficial) and the new capabilities. They should optimize the new capabilities (in terms of content, covering rate, etc.), respecting the standard requirements for real applications. Regarding the costs of the application, $TiO_2$ nanoparticles are still quite expensive because their industrial production is still quite low. However, it is predicted that the cost will decrease significantly in the next years. Since there is a lack of information about the cost analysis, this approach could be evaluated by the researchers in further studies.

It can be concluded that photocatalytic road pavements are very important since they can be used in places where there are high emissions of pollutants (typically in areas with a high density of urban mesh), since these modified road structures are characterized by having a high surface area and, therefore, present a great potential to effectively promote photocatalytic reactions with the surrounding pollutants, thus contributing to the improvement of health conditions of populations that live in those urban centers. However, in real contexts (functionalized pavements with great extension, tens or hundreds of kilometers) the existing experience on application techniques is unsatisfactory, and some problems mostly associated with the surface immobilization of the semiconductor materials need further study.

**Author Contributions:** I.R.S. wrote the paper; S.L.J. wrote the Section Basic Principles of Photocatalysis; E.F., J.O.C. and M.F.M.C. reviewed the paper, improving the content quality, and polished the grammar. Conceptualization, I.R.S. and S.L.J.; Methodology, I.R.S., S.L.J. and J.O.C.; Formal Analysis, I.R.S. and S.L.J.; Investigation, I.R.S., S.L.J. and J.O.C.; Writing—Original Draft Preparation, I.R.S. and S.L.J.; Writing—Review and Editing, J.O.C., M.F.M.C. and E.F.; Visualization, I.R., J.O.C.; Supervision, J.O.C., M.F.M.C. and E.F.; Project Administration, J.O.C. and E.F.; Funding Acquisition, J.O.C., M.F.M.C. and E.F.

**Funding:** This work was partially financed by FCT—Fundação para a Ciência e a Tecnologia—under the projects of the Strategic Funding UID/FIS/04650/2019, Nanobased concepts for Innovative and Eco-sustainable constructive material's surfaces PTDC/FIS/120412/2010, and PEst-OE/ECI/UI4047/2019. Also, the first author would like to acknowledge FCT for the PhD scholarship (SFRH/BD/137421/2018).

**Conflicts of Interest:** The authors declare no conflict of interest.

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
