# Peer review of "Smart, Photocatalytic and Self-Cleaning Asphalt Mixtures: A Literature Review"

_coatings, doi:10.3390/coatings9110696_

Round 1

Reviewer 1 Report

This is a review article. Authors have listed some of the prominent research article in the field. My recommendations are as follows:

  1. Check and revise English language through out the paper.
  2. Add references for example line 31.
  3. Language used is non technical and difficult to read. Unnecessary details and repetitions could be removed.
  4. I suggest to give details of the things that are in line with the objective and scope of the study.  Clearly state objective/scope of the research.
  5. Google hits are non technical details.
  6. Page 6-13 have no sections or subsections which makes the article extremely difficult to comprehend.

Author Response

1. Review CommentsThis is a review article. Authors have listed some of the prominent research article in the field. My recommendations are as follows: Check and revise English language through out the paper.

Thank you for your revision. Your comments have substantially improved the manuscript. English language was revised.

2. Add references for example line 31.

One reference was added to this line: Hashimoto;, K.; Iire, H.; Fujishima, A. TiO2 Photocatalysis : A Historical Overview and Future Prospects. Jpn. J. Appl. Phys. 2005, 44.

3. Language used is non technical and difficult to read. Unnecessary details and repetitions could be removed.

Some information has been removed in order to provide clearer reading. Language was also revised.

4. I suggest to give details of the things that are in line with the objective and scope of the study.  Clearly state objective/scope of the research.

Taking into account the objective of this manuscript, the paragraph was added in page 2 (lines 69-80):

“The most important new functions imparted to asphalt mixtures are the photocatalytic and self-cleaning properties. Therefore, the main goal of this article is to present a review about the functionalization processes performed over asphalt mixtures and also the discussion of their most relevant features. Firstly, general characteristics of titanium dioxide semiconductor material and essential principles of photocatalysis will be discussed in Chapters 2 and 3, respectively. Afterwards, some applications related with civil engineering materials will be presented in Chapter 4. Concerning the particular applications for asphalt mixtures, Chapter 5 presents the main used materials and the corresponding published topics, application methods and its benefits and limitations, some surface modifications as a strategy to improve photocatalytic efficiency such as TiO2 metal doping or coupled semiconductor photocatalysts (e.g. g-C3N4–TiO2 or CdS–TiO2 systems), analysis of results from literature, modeling of photocatalytic pavements, impact on the essential characteristics of asphalt mixtures and application to experimental and real road sections and also their related costs”.

5. Google hits are non technical details.

We have used Google Scholar in order to show the importance and the innovation of this subject. This is one of the most important search platforms for research. In this analysis, the number of occurrences of some keywords was quantified, and it was concluded that photocatalysis applied to this research area is very innovative and raises great interest for the researchers.

6. Page 6-13 have no sections or subsections which makes the article extremely difficult to comprehend.

We have divided the Chapter “Photocatalytic Asphalt Pavements into 6 subsections”:

5.1. Materials and Techniques for Application of Photocatalytic Effect on Asphalt Mixtures

5.2. Testing, Results Analysis of Photocatalytic Asphalt Mixtures

5.3. Photocatalytic Asphalt Mixtures Modeling

5.4. Mechanical and Surface Impacts and Design of Photocatalytic Asphalt Mixtures

5.5. Photocatalytic Applications in Real Context or Small Road Sections

5.6. Cost Analysis

Reviewer 2 Report

In this manuscript, the authors carry out a Review about the use of photocatalytic materials on asphalt mixtures for self-cleaning applications. 
1. This is a very interesting topic, considering not only the fundamental aspects of photocatytic degradation with TiO2 materials, which has been extensively studied, but also the realistic application of such materials into asphalts and paintings, closing the gap between the technological part and the market. Although the authors have made a proper collection of some research works, including some real application studies, I consider the manuscript requires an extensive edition and reorganization. Specially in the first part, where the authors aim to describe the principles behind the photocatalytic degradation, many paragraphs seems rather repetitive. As examples, the paragraph after line 96 repeats what the authors have previously described, about superhydrophilicity. The same as lines 248-251. 

2. Therefore, besides the extensive english revision required, I would recommend the author to re-organize the manuscript. Moreover, a better separation into different subsections would be helpful for the reader. For instance, the section describing the realistic application examples could be included in a separate sub-section.

3. Also, I have some other comments: Lines 147-151, although this is true, sometimes the defects might become recombination centers and be detrimental to the photocatalytic performance.

4. Figure 1 is not specific for TiO2 as indicated in line 123. This scheme is general for any semiconductor.

5. What do the authors mean by "hv" of the semiconductor? It is not a property of the semiconductor. Please keep it in mind for all the paragraph (lines 159-167).

6. Are Figures 4 and 5 a reproduction of other publications?

7. For lines 541-543, regarding the TiO2 cost, please provide some values and references.

Author Response

1. Review CommentsIn this manuscript, the authors carry out a Review about the use of photocatalytic materials on asphalt mixtures for self-cleaning applications. This is a very interesting topic, considering not only the fundamental aspects of photocatytic degradation with TiO2 materials, which has been extensively studied, but also the realistic application of such materials into asphalts and paintings, closing the gap between the technological part and the market. Although the authors have made a proper collection of some research works, including some real application studies, I consider the manuscript requires an extensive edition and reorganization. Specially in the first part, where the authors aim to describe the principles behind the photocatalytic degradation, many paragraphs seems rather repetitive. As examples, the paragraph after line 96 repeats what the authors have previously described, about superhydrophilicity. The same as lines 248-251.  

Thank you for your very nice comments. Surely, your revision has improved our manuscript. We have reorganized the manuscript mainly taking into account the Chapter 5. We created 6 subsections: 5.1. Materials and Techniques for Application of Photocatalytic Effect on Asphalt Mixtures5.2. Testing, Results Analysis of Photocatalytic Asphalt Mixtures5.3. Photocatalytic Asphalt Mixtures Modeling5.4. Mechanical and Surface Impacts and Design of Photocatalytic Asphalt Mixtures5.5. Photocatalytic Applications in Real Context or Small Road Sections5.6. Cost Analysis We tried to change the text in order to avoid repetitive parts in the manuscript. Considering the paragraphs related with the superhydrophilicity, we have combined the following information: 

“Besides the photocatalytic properties, TiO2 presents a superhydrophilic surface under photoinduction. TiO2 surface can change between a less photocatalytic and more superhydrophilic state depending on the chemical composition and processing techniques. When this semiconductor material is irradiated by UV light, the water adsorbed on the surface of TiO2 spreads forming a thin film, which is related to the reconstruction of hydroxyl groups under UV-light irradiation. Molecular oxygen capture the photo-excited electrons and holes diffuse to the TiO2 surface where they can be trapped by surface lattice oxygen’s, as suggested by Nakamura et al. [Nakamura and Y. Nakato. Primary intermediates of oxygen photoevolution reaction on TiO2 (rutile) particles, revealed by in situ FTIR absorption and photoluminescence measurements. J. Am. Chem. Soc., 126 (4) (2004) 1290-1298].

Hole trapping weakens the energy between the Ti atoms and lattice oxygen. Another adsorbed water molecule breaks this bond, forming a new hydroxyl group. The successive dissociative adsorption of water induces the trapping of these hydroxyl species leading to the photogeneration of a hydrophilic domain (size of about 10 nm). Given the unstable state of this surface, the bound photogenerated hydroxyl groups desorb gradually and the surface returns to its initial state presenting less hydrophilic behavior [11,23]. TiO2 material’s surface with flat or rough texture can present hydrophobic or superhydrophobic behavior, when coated with hydrophobic materials such as, for example, some silane compounds. Moreover, TiO2 surfaces can also present a superhydrophilic state, which in general, is characterized by a water contact angle less than 5°, when irradiated with UV-light [15]. ”

 We have removed the information about the road pavements at the end of the Chapter 4 (lines 248-251 from the first version of the manuscript) since that information is now described in Chapter 5. 

2. Therefore, besides the extensive english revision required, I would recommend the author to re-organize the manuscript. Moreover, a better separation into different subsections would be helpful for the reader. For instance, the section describing the realistic application examples could be included in a separate sub-section. 

We have revised the English Language. As mentioned before, we dived the Chapter 5 into 6 subsections. 

3. Also, I have some other comments:- Lines 147-151, although this is true, sometimes the defects might become recombination centers and be detrimental to the photocatalytic performance.

The reviewer's comment is pertinent. In fact, as the reviewer points out, in preparation of some colloidal and polycrystalline photocatalysts, ideal crystal lattices of the semiconductors are not formed. Instead, surface and bulk defects occur quite regularly during the photocatalysts preparation process. The nature of surface defect sites depends on the semiconductor's preparation method. As a particular example about the role of surface traps, is the preparation of CdS colloids obtained by the addition of H2S to a cadmium salt solution, which generate surface defect sites promoting radiation-less recombination of charge carriers in this semiconductor systemFurthermore, if a surface modification is performed by the addition of excess Cd2+ ions and with a proper pH adjustment toward a basic solution, then it is induced the blocking of charge carriers trap sites, which also promotes radiation-less recombination of electron-hole pairs across the semiconductor bulk band gap, therefore being detrimental to the photocatalytic performance [R1, R2].These considerations are now appropriately inserted in the new version of the manuscript. [R1] Lubomir Spanhel, Markus Haase, Horst Weller, Arnim Henglein. A. Photochemistry of colloidal semiconductors. Photochemistry of colloidal semiconductors. 20. Surface modification and stability of strong luminescing CdS particles. J. Am. Chem. Soc., 109 (19) (1987) 5649-5655. [R2] Amy L. Linsebigler, Guangquan Lu, John T. Yates, Jr. Photocatalysis on TiO2 surfaces: principles, mechanisms, and selected results. Chem. Rev. 95 (3) (1995) 735-758. 

4. Figure 1 is not specific for TiO2 as indicated in line 123. This scheme is general for any semiconductor. 

We have changed this information. Now the scheme is given for any semiconductor material. 

5. What do the authors mean by "hv" of the semiconductor? It is not a property of the semiconductor. Please keep it in mind for all the paragraph (lines 159-167). 

All the Chapter 3 was revised. Now Eg and hv are well described in page 3: “In terms of energy, semiconductors possess an emptiness energy space region where there are no energy levels available to promote recombination of electron-hole pairs generated by photoactivation in the solid semiconductor.The empty energy region is the so-called band gap energy, which extends from the top of the filled valence band to the bottom of the empty conduction band of the semiconductor material. Since photoexcitation occurs via the band gap, the generated electron-hole pairs have an enough lifetime, in the nanosecond range, to promote charge transfer to adsorbed species on the semiconductor surface (from the solution or from the gas phase in contact) and participate in redox processes [25]”. The interaction of the photogenerated electron-hole pairs with chemical species adsorbed to the semiconductor surface generally occurs via two main pathways. At the semiconductor surface, the electrons can be transferred to acceptor species, thus being reduced (usually the electron acceptor is oxygen in an aerated solution); in turn, a hole migrates to the semiconductor surface where it can be combined with donor species, which become oxidized. The initial step of semiconductor photocatalysis occurs when a photon with an equal (or greater) energy (hv) than the Eg is absorbed by the semiconductor material. 

6. Are Figures 4 and 5 a reproduction of other publications? 

Figure 4 is from our previous work (this figure was not published before), but Figure 5 is not. It was described into the text in page 9, but we have included the reference above the Figure. Figure 5: Results of photocatalytic tests: a) Variation (over time) of gas concentration, indicating its photodegradation [63] and b) absorbance variation (over time) of the dye's aqueous solution, indicating its photodegradation [7]. 

7. For lines 541-543, regarding the TiO2 cost, please provide some values and references

Previously, we have discussed the application costs referred to the spraying coating technique. Now we have included some additional information about the raw TiO2 nanoparticles: “As a typical example for the costs of the raw TiO2 nanoparticles, on can highlight the case of Evonik Aeroxide P-25, which present a cost of about USD 45 per kg [88].” 

Reviewer 3 Report

The authors present a work on the smart, photocatalytic and self-cleaning asphalt mixtures: a literature review. The subject of the authors work is an important significant issue in structural engineering and materials, and such an attempt is of great interest. 

1. The authors of the article have made a very interesting and extensive review of the literature. It is worth noting that many literature items are very new.

2. However the paper, in its present form, requires some substantial modifications in order to justify its publication in an International Journal such as Coatings. I think that the following points should be further elaborated by the authors: the descriptions of the graphs shown in Figure 5b should be more legible (please enlarge them);the pictures shown in Figure 6 are blurred. I suggest you enlarge the photos.

3. I have no substantive comments. I think the article is very well written.

Author Response

REVIEWER 3:

Review Comments 

1. The authors present a work on the smart, photocatalytic and self-cleaning asphalt mixtures: a literature review. The subject of the authors work is an important significant issue in structural engineering and materials, and such an attempt is of great interest.  The authors of the article have made a very interesting and extensive review of the literature. It is worth noting that many literature items are very new. 

Thank you for your amazing comments.

2. However the paper, in its present form, requires some substantial modifications in order to justify its publication in an International Journal such as Coatings. I think that the following points should be further elaborated by the authors:the descriptions of the graphs shown in Figure 5b should be more legible (please enlarge them), the pictures shown in Figure 6 are blurred. I suggest you enlarge the photos. 

We have enlarged the Figures 4 and 5. Since Figure 6 was reproduced from another article, we tried to use the best quality taking into account this limitation. If we enlarge Figure 6, it will greatly lose its quality. 

3. I have no substantive comments. I think the article is very well written. 

Thank you!! The comments improved the manuscript a lot.

Round 2

Reviewer 1 Report

The paper has been revised and reads a lot better. However, still English language should be rechecked.My recommendations are as follows:

1. Change Chapters to sections.

2. Remove line 33.

3. The authors may reduce part of the initial chapters (up to page 8) that are not relevant  to their research objective (which is about pavements). If these information are necessary please show it relevance to pavements.

4. Why do the authors titled the paper as smart and self-cleansing? please add a justification somewhere in the paper.

Author Response

The paper has been revised and reads a lot better. However, still English language should be rechecked. My recommendations are as follows:

1. Change Chapters to sections.

 The name chapter was replaced to section in all the manuscript.

2. Remove line 33.

The line 33 was removed.

3. The authors may reduce part of the initial chapters (up to page 8) that are not relevant  to their research objective (which is about pavements). If these information are necessary please show it relevance to pavements.

Let us disagree with the reviewer. The introduction and the information about the General Aspects and Characteristics of Titanium Dioxide, Basic Principles of Photocatalysis and General Application for Civil Engineering Materials are essential. In the introduction section, we take into account the first approaches of the photocatalysis, the description of smart materials and the importance of the application of photocalysis on asphalt mixtures. Regarding the Section General Aspects and Characteristics of Titanium Dioxide, we show the most important characteristics of TiO2 in order to ensure that the readers better understand what type of semiconductor material is being inserted in asphalt mixture in order to promote the photocatalytic process; the reason why it is mostly used anatase and P25 nanoparticles. Considering the Basic Principles of Photocatalysis, we explain how the photocatalysis occurs, also showing the reaction of NO2 and organic compounds, the main characterizations to evaluate the energy band gap and the importance of TiO2 doping. In addition, it is probable that this article will be addressed mostly to Civil Engineering researchers. Thus, it is essential to show the characteristics, principles and the main chemical reactions related to photocatalysis. The Chapter General Application for Civil Engineering Materials was written in order to show the importance of the application of semiconductor on different Civil Engineering materials. Usually, the new materials are applied in other areas of Civil Engineering, and then it is inserted on Road Engineering.

4. Why do the authors titled the paper as smart and self-cleansing? please add a justification somewhere in the paper.

 Some authors describe the smart material as a “a material with properties that differ from the conventional ones, which can react to external stimulus complying with those requirements, such as stress and temperature. These properties can be self-healing, self-sensing, electrically conductive, electromagnetic and thermal. A smart and multifunctional material is accomplished by properly composition design, special processing, introduction of new functional components, or via the modification of the microstructure from the conventional one [5]”. The smart concretes are classified by its smartness, mechanical, electrical, optical, electromagnetic wave/radiation shielding/absorbing, water-related, and energy-harvesting functions. The photocatalytic is related to the optical functions. So, because of these reason, the photocatalytic asphalt mixtures are smart.

We explain in page two the self-cleaning property of these smart materials: “Since in the presence of light irradiation, photocatalytic materials can degrade organic pollutants (such as oils and greases) adsorbed to their surface, these materials can also be classified as self-cleaning materials, a very important property in road engineering applications, because the self-cleaning function can contribute to a significant reduction of car accidents on oil-spilled areas”. Thus, these materials are classified as self-cleaning.

Reviewer 2 Report

In the present version, the authors have improved the structure of the manuscript, making it more easily readable and better organized. As for the previous version, I consider it is an interesting topic closing the gap between lab-scale research and technological and commercial applications. In that sense, this review makes an interesting compilation on studies carried out under realistic conditions and some real examples of self-cleaning paintings and asphalts. Also, some misleading concepts when describing general aspect about heterogeneous photocatalysis have been corrected.
Some minor comments: 

1. Please correct last sentences on the Abstract.

2. In line 86, “Typically, TiO2 appears in four different geometric/dimensional shapes (0 to 3)”, it sounds as a property associated to TiO2, but a lot of materials can be synthesized for having these dimensional shapes.

3. Just as a comment, I personally don’t find the paragraph between lines 324-334 regarding the number of articles has any significant contribution to the manuscript.

Author Response

In the present version, the authors have improved the structure of the manuscript, making it more easily readable and better organized. As for the previous version, I consider it is an interesting topic closing the gap between lab-scale research and technological and commercial applications. In that sense, this review makes an interesting compilation on studies carried out under realistic conditions and some real examples of self-cleaning paintings and asphalts. Also, some misleading concepts when describing general aspect about heterogeneous photocatalysis have been corrected.

 Some minor comments:

1. Please correct last sentences on the Abstract. 

Thank you. We corrected this sentence.

2. In line 86, “Typically, TiO2 appears in four different geometric/dimensional shapes (0 to 3)”, it sounds as a property associated to TiO2, but a lot of materials can be synthesized for having these dimensional shapes.

We changed the sentence: “As with different materials, TiO2 is produced in four different geometric/dimensional shapes (0 to 3)”.

3. Just as a comment, I personally don’t find the paragraph between lines 324-334 regarding the number of articles has any significant contribution to the manuscript.

In this paragraph, it is shown the innovation and interest of the researches considering the application of photocatalysis on asphalt mixtures. In this search, the number of occurrences about mechanical properties of asphalt mixtures was 65% while it was 1% for photocatalytic asphalt mixtures. Nevertheless, this number increased from 2006 to 2018 (2 to 40 occurrences). Thus, the interest of the research groups for this subject is increasing but it is still quite low comparing to the main topic (mechanical behavior of asphalt mixtures).